# Early Intra-Abdominal Bacterial Infections after Orthotopic Liver Transplantation: A Narrative Review for Clinicians

**DOI:** 10.3390/antibiotics12081316

**Published:** 2023-08-15

**Authors:** Riccardo Taddei, Niccolò Riccardi, Giusy Tiseo, Valentina Galfo, Giandomenico Biancofiore

**Affiliations:** 1Division of Transplant Anesthesia and Critical Care, Department of Anesthesia, Azienda Ospedaliero Universitaria Pisana, University of Pisa, 56124 Pisa, Italy; g.biancofiore@med.unipi.it; 2Infectious Diseases Unit, Department of Clinical and Experimental Medicine, Azienda Ospedaliero Universitaria Pisana, University of Pisa, 56124 Pisa, Italy; niccolo.riccardi@yahoo.it (N.R.); tiseogiusy@gmail.com (G.T.); vale.galfo@gmail.com (V.G.)

**Keywords:** intra-abdominal infections, surgical site infections, liver transplantation, multidrug resistance, bacterial infections

## Abstract

Despite recent advances in the transplant field, infectious complications after orthotopic liver transplantation (OLT) are major causes of morbidity and mortality. Bacterial intra-abdominal infections (IAIs) are predominant during the first month post-transplantation and affect patient and graft survival. Recently, the emergence of multidrug resistant bacteria has generated great concern in OLT patients. We performed this narrative review of the literature in order to propose a “ready-to-use” flowchart for reasoned empirical antibiotic therapy in the case of suspected post-OLT IAIs. The review was ultimately organized into four sections: “Epidemiology and predisposing factors for IAI”; “Surgical-site infections and perioperative prophylaxis”; “MDRO colonization and infections”; and “Reasoned-empirical antibiotic therapy in early intra-abdominal infections post OLT and source control”. Multidisciplinary teamwork is warranted to individualize strategies for the prevention and treatment of IAIs in OLT recipients, taking into account each patient’s risk factors, the surgical characteristics, and the local bacterial epidemiology.

## 1. Introduction

Orthotopic liver transplantation (OLT) is an effective treatment for end-stage liver diseases. Recipient survival has risen thanks to the improvement in surgical techniques, perioperative management, and immunosuppressive regimens, with the 5-year survival rate currently exceeding 70% [1]. Among OLT recipients, about one half of the deaths occurring within 1 year after the transplant are caused by infections [2]. Bacterial infections are responsible for up to 70% of all infectious episodes, and about half of them occur within 2 weeks of the surgery [3]. The frequency, type of infection, and specific microorganisms involved generally follow a predictable time to onset. In the first 30 days after OLT, infections are usually linked to the surgical procedure itself or are healthcare-associated. Bacterial infections are prevalent at this stage [4,5]. Between one and six months after OLT, bacterial infections are less prevalent, and because of the increased burden of immune suppression, opportunistic infections are more likely to develop. These predominantly include Cytomegalovirus (CMV), *Pneumocystis jirovecii*, fungi, and *Toxoplasma gondii* [3]. Six months after OLT, the net state of immunosuppression tends to decrease, as does the risk of opportunistic infections. Transplant recipients remain at a high risk for community-acquired infections, and recurrent cholangitis may develop in patients with chronic graft complications such as biliary strictures [5,6]. Reactivation of viral infections may also manifest in the longer term.

Infections in this patient population are notoriously difficult to diagnose because the usual signs and symptoms of infection (fever, leukocytosis, and biomarkers levels) may be masked or absent due to the net state of immunosuppression.

In this narrative review, we provide a comprehensive overview of early bacterial intra-abdominal infections (IAIs) after OLT, with the aim to develop a practical therapeutic tool useful for physicians in their clinical practice.

## 2. Results

### 2.1. Epidemiology and Predisposing Factors for IAI

Bacterial infections, especially caused by hospital-acquired pathogens, are prevalent in the early phase after OLT [3]. During this phase, the most important sites of infection are the abdomen, the surgical wound, the bloodstream, and the urinary and respiratory systems [6].

IAIs account for up to 50% of early bacterial infections following OLT and include peritonitis, cholangitis, and more rarely, hepatic abscesses [7,8]. IAIs often occur at the surgical site and can be associated with long and complicated surgeries or due to persistent ascites’ superinfection [9]. IAIs are usually polymicrobial, mainly caused by enteric pathogens. *Enterococci* are the most commonly isolated species, followed by Enterobacterales, *Pseudomonas* spp., *Staphylococcus aureus*, and anaerobic germs [7,9]. Risk factors for IAIs in the early postoperative phase include donor infections, surgical complications, and nonsurgical (host) factors.

The risk of donor-derived infections should be stratified through a careful review of the donor’s medical and social history. Particular attention should be taken to the travel history to identify donors at risk of endemic infections. A thorough physical evaluation of the donor’s body by the organ procuring team should be conducted, with particular attention to the explanted liver and related vessels, abscesses, ulcers, skin rash, and lymphadenopathy. Decisions regarding the use of organs from donors with suspected bacterial infections should be based on the urgency of the transplantation for the recipient, the availability of alternative organs, and recipient informed consent [10]. Organs known to be infected with pathogens likely to be transmitted to the recipient should be discarded unless there is confidence that the recipient can safely be treated post OLT. In this setting, each case should be discussed individually, and the involvement of the Transplant Infectious Diseases team is strongly encouraged to develop a postoperative management plan [10].

Surgical complications are significantly associated with an increased risk of surgical site infections (SSIs) and IAIs [9]. The transfusion of multiple packed red blood cells plays an immunosuppressive role and implies a prolonged and more complex procedure, associated with an increased risk of postoperative infections [5,11]. Since the hepatic artery is the major blood supply to the bile duct in OLT recipients [12], hepatic artery thrombosis or stenosis, which reduce the oxygen delivery to the biliary tract, are associated with both short- and long-term biliary complications including cholangitis, bilomas, and liver abscesses [6]. Roux-en-Y choledochojejunostomy, compared to duct-to-duct biliary anastomosis, is associated with an increased risk of intestinal perforation, stricture, leakage, bleeding at the jejuno–jejunostomy site, and ascending cholangitis [13]. Biliary strictures that account for up to one third of all biliary complications within the first year after transplant, predispose to recurrent cholangitis [14]. Bile leaks develop in 2–25% of patients after OLT and are associated with chemical peritonitis, which may result in a secondary infection [13].

Nonsurgical factors include a Model for End-Stage Liver Disease (MELD) score greater than 30, renal replacement therapy, a prolonged stay in the intensive care unit, mechanical ventilation, indwelling vascular and urinary catheters, older age, and profound immunosuppression [5,15].

Bilomas occur in around 11% of patients after OLT and are formed by bile extravasation into the liver or abdominal cavity due to biliary ischemia resulting in bile duct injury. These masses are commonly associated with hepatic artery thrombosis. If the bile is already colonized by bacteria, bilomas become colonized as well, though they can also develop a secondary infection [16,17]. The diagnosis of bilomas is made with the aid of imaging techniques such as abdominal ultrasound, computed tomography (CT), or magnetic resonance imaging (MRI), as their signs and symptoms are usually nonspecific. In a study of 57 OLT patients with biloma by Safdar et al. [7], the patients were asymptomatic in 35% of cases, while fever and abdominal pain were the most observed symptoms in the remaining ones. Elevation of hepatic enzymes was the most frequent laboratory test alteration detected (79% of cases), and jaundice was rare (28%). The most commonly isolated bacteria were *Enterococcus* spp., followed by Gram-negative bacilli. The occurrence of infected bilomas is associated with an increased hospital length of stay (LOS), graft loss, and mortality [7].

In a retrospective study of 1100 OLTs at the Medical University of Warsaw, hepatic abscesses occurred in 1.4% of cases [18]. The signs and symptoms observed, as well as the diagnostic workup, did not differ from bilomas. The micro-organisms responsible for the infection, on the other hand, were mainly Gram-negative bacilli, followed by *Staphylococcus aureus* and *Enterococcus* spp. Polymicrobial abscesses were relatively common (up to 40% of cases). The mortality associated with hepatic abscesses ranged from 33% to 42% [18].

As a further complication, ascending cholangitis accounts for 6–7% of infections in OLT recipients. Its presenting symptoms are nonspecific, while the most frequent sign is cholestasis. CT scan and MRI are valuable diagnostic aids to detect predisposing factors, such as bile duct strictures [19].

Finally, peritonitis after liver transplantation occurs in up to 10% of OLT recipients. In a retrospective study of 950 liver transplants developing postoperative peritonitis, more than half of the cases occurred in association with surgical complications, such as intra-abdominal bleeding, biliary leak or stricture, bowel perforation, anastomotic leak, and wound infections [20]. In most of the cases, the ascitic fluid samples revealed polymicrobial infections, with a high prevalence (up to 76%) of enteric flora and multidrug-resistant organisms (MDROs).

### 2.2. Surgical Site Infections and Perioperative Prophylaxis

Although there has been progress in infection control practices, SSIs remain one of the most common healthcare-associated infections and a significant cause of morbidity and mortality in postsurgical patients [21]. Liver transplant recipients have an increased rate of SSIs compared with other solid organ transplants, ranging from 10% to 37%, with SSIs representing a significant cause of morbidity, prolonged hospitalization, graft loss, and mortality within 1 year [20,22,23].

The Centre for Disease Control and Prevention (CDC) categorizes SSIs according to the involved anatomical space: (1) superficial incision infections, (2) deep wound infections, or (3) organ/organ space infections. The infection must occur within 30 days of surgery or within 90 days if a mesh is used for wound closure [24,25] to be categorized as an SSI.

While the updated 2017 CDC guidelines and 2016 World Health Organization (WHO) guidelines for prevention of SSIs specifically address the risk factors and underline prevention as a way to methodically reduce the infection rates in the perioperative period, they do not specifically address the topic of solid organ transplant [26,27]. Liver transplant recipients represent a unique high-risk population due to the complexity of the surgical technique, the risk of blood loss, and immunosuppression. The risk factors for SSIs include:(a)Donor factors such as donor infection;(b)Host factors such as a profound state of immunosuppression, prolonged intensive care unit stay, antibiotic use in the 3–4 months before surgery, diabetes, high MELD score, hemochromatosis, ascites, obesity, prior hepatobiliary surgery, prior liver or renal transplantation, postoperative acute rejection, or the need for renal replacement therapy (RRT);(c)Surgical factors, such as a prolonged duration of surgery (>8–12 h), choledocho–jejunal anastomosis, and an intraoperative transfusion of >4 units of blood [22,28].

Local signs and symptoms of infection, such as pain, swelling, or purulent drainage, together with fever and leukocytosis, represent the most common elements leading to an SSI diagnosis [25]. However, due to the immunosuppression needed to avoid organ rejection, these may not always be present. The diagnosis of SSI, therefore, requires a thorough evaluation of the patient including clinical examination, laboratory tests, and diagnostic imaging. Surgical exploration or percutaneous drainage may be indicated to control the SSI, together with appropriate empirical antibiotic therapy, which should be discussed on an individual basis.

Among the prevention strategies for reducing SSIs, surgical antibacterial prophylaxis (SAP) is the standard of care for nearly all surgical procedures, including OLT. This minimizes the risk of SSIs by decreasing the bacterial burden at the surgical site, thereby optimizing the environment for healing. Other necessary steps to limit SSIs include minimizing the surgical operative time, optimizing sterile techniques, and maintaining a good perioperative management of patient comorbidities, as well as oxygenation, volemia, glucose, and temperature regulation [26]. Multiple international guidelines have been published for antibiotic selection and dosing, but there is no consensus specific for the transplant population. Therefore, SAP regimens and durations vary widely across different transplant centers [29].

Organisms most commonly causing SSIs in OLT recipients are Gram-negative bacilli (predominantly Enterobacterales), *Enterococcus* spp. and *Staphylococcus aureus* [9,30,31,32]. As a consequence, regimens covering these bacterial species should be considered. Garcia Prado et al., in 2008, found no difference in the rate of SSIs when comparing amoxicillin–clavulanate prophylaxis with the use of a first-generation cephalosporin [33]. However, a larger study of 1222 OLT patients found, using univariate analysis, that the use of cefazolin was associated with a significantly higher risk for SSI compared with other regimens, suggesting that a broader antibiotic coverage, with activity against *Enterococcus* spp., may be beneficial. However, no difference was observed after controlling for confounding factors [30].

Considering all the findings above, the IDSA/ASHP/SIS/SHEA guidelines and the ERAS4OLT.org Working Group expert panel recommended the administration of a third-generation cephalosporin plus ampicillin, piperacillin/tazobactam, or amoxicillin/clavulanate [34,35]. According to the recommendation, SAP should be administered within 120 min prior to the incision, to ensure adequate concentrations in serum and tissues, and the antibiotic should be re-administered based on its half-life during the surgical procedure [27].

Regarding the SAP duration, a single center pilot randomized control trial, published in 2019, enrolled 102 patients divided into two groups, one receiving only intraoperative doses of prophylactic piperacillin/tazobactam and one receiving an extended regimen for 72 h. The study showed no difference between the two groups in terms of the rates of SSIs (19% in the short-course and 27% in the long-course group), overall infections, intensive care unit (ICU) length of stay, hospital length of stay, and rate of readmission or reoperation [36]. Another recent study including 44 patients compared different antibiotic regimens and durations, including ampicillin/sulbactam and piperacillin/tazobactam, and found no increase in post-transplant infections even with a shorter prophylaxis regimen (from 72 to 24 h) [37]. For the general surgery population, both the CDC and WHO recommend cessation of antibiotics upon closure of the surgical site [26,27]. A recently published systematic review of the literature and expert panel recommendations by the ERAS4OLT.org Working Group recommend a length of SAP not exceeding 24 h in patients undergoing OLT [35]. It should be noted, however, that so far no adequately powered randomized clinical trials have addressed this topic in a liver transplant population.

Selective intestinal decontamination (SID), either with topical antibiotics in suspension formulations or using systemic antibiotics with little anaerobicidal activity, aim to eradicate potential pathogenic germs from the digestive tract. In particular, this strategy targets aerobic Gram-negative bacilli, *Enterococcus* spp. and *S*. *aureus*, without significantly affecting the normal commensal anaerobic flora [38]. Multiple studies evaluating the role of SID in liver transplant patients showed controversial results, probably due to the significant heterogeneity in methodology: the SID regimens that were used, the timing of the administration, and the duration of treatment were significantly different across studies [38]. To date, there has been no conclusive benefit demonstrated from the use of SID in liver transplantation patients, and this intervention is not currently recommended, especially given the potential risk of selecting resistant microorganisms [22,39,40,41].

### 2.3. MDRO Colonization and Infections

Patients with infections by MDROs have increased mortality compared to patients with infections caused by susceptible organisms [42]. An increasing number of infections due to MDRO has been documented among OLT recipients, with a highly variable number of cases among different transplant centers [43]. Importantly, solid organ transplant recipients represent a peculiar patient population with a high risk of mortality and poor outcome after MDRO infections [44]. As such, this population should be carefully evaluated and is likely in need of a specific approach on an individual basis.

We could not find any study specifically addressing the rate and significance of IAI due to MDROs in liver transplant recipients. However, according to Kawecki et al., the overall predominantly isolated species are vancomycin-resistant *Enterococci* (VRE) and extended-spectrum beta lactamases (ESBL) producing Enterobacterales [32]. Most of the studies in the literature identify preoperative colonization as a significant risk factor for the development of subsequent post-OLT infections [45]. Thus, a thorough preoperative screening of the transplant candidate, including information about the past medical history, previous hospitalization, surgical interventions, and immunosuppression, is crucial. Data about microbiological colonization and the history of recent antimicrobial exposure should be obtained before OLT [6]. The importance of knowing about a potential intestinal colonization has been already described in hospitalized patients, including transplant recipients. This knowledge, in fact, is critical as it allows the adoption of control measures as a precaution, including tailored antibiotic therapy (when needed) based on the specific rectal colonizing organisms [46].

Nasal or rectal colonization by methicillin-resistant *S. aureus* (MRSA) in transplant recipients has been found to occur in 6.7% to 22% of cases [45]. Identification of MRSA carriers has important implications for empirical therapy, and the eradication of MRSA may seem a valuable option for limiting *S. aureus* infections. However, current decolonization strategies using topical local mupirocin have given controversial results. In a study conducted by Paterson et al. [47] on OLT recipients and candidates, decolonization of *S. aureus* carriers with nasal mupirocin resulted in a 37% rate of recolonization, with an increased prevalence of MRSA among the re-colonized patients. Therefore, the American Society of Transplantation Infectious Diseases Community of Practice recommends that the transplant centers should evaluate the local prevalence of MRSA colonization and wound infections and tailor the use of nasal decolonization to their individual population [25].

Vancomycin-resistant *Enterococci* (VRE), especially *E. faecium*, represents a concern in OLT recipients. Rectal colonization is reported in 3–55% of cases, with rates of infection among colonized patients ranging between 12% and 32% [5]. In a study of 375 liver transplant candidates, VRE colonization was associated with greater morbidity but not greater mortality than non-colonized candidates [48]. Contact isolation of these patients is currently recommended, while there is no evidence to guide the decision on whether the perioperative prophylaxis should be altered based on the recipient VRE colonization [5]. Other risk factors for VRE infections include procedures related to the biliary tract, surgical re-exploration, long hospital stay, and prior antibiotic use [45].

Rates of infections in OLT patients by Gram-negative MDRO are constantly increasing worldwide, reaching up to 50% in some centers, and are recognized as a significant cause of increased mortality compared with infections caused by non-resistant bacteria [49]. Infections caused by ESBL producing Enterobacterales in OLT have rates between 6% and 40% (in endemic countries) across transplant centers. The risk of infection increases in patients already colonized pre-transplant, as well as in those with advanced liver disease and requiring surgical revision [5].

Carbapenem-resistant Enterobacterales, in particular carbapenem-resistant *K. pneumoniae* (CRKP), has emerged as a major concern for immunocompromised patients, associated with a significant mortality [5,50]. Pre- and post-transplant colonization has emerged as an important factor associated with CRKP infection, together with prior hospitalization, higher MELD at OLT, prior antibiotic exposure, and time spent in the ICU [51]. Importantly, the production of different carbapenemases may be responsible for carbapenem resistance among CRKP. Therefore, ascertaining which types of carbapenemases (such as New Delhi metallo-beta-lactamases [NDM], Verona imipenamases [VIM], KPC, and OXA-48 like) are expressed by the colonizing bacteria has several important implications in clinical practice [52]. The role of rapid testing useful to identify specific carbapenemase families has been underlined and emphasized by recently published guidelines [53].

There are currently no guidelines specifically addressing the management of MDRO infections, in particular IAIs, in OLT recipients. Although there has been some debate on the use of a specific SAP regimen for MDRO colonized candidates, the specific approach to use for these patients should be individualized according to the local epidemiology. The involvement of the local Antimicrobial Stewardship Team and Transplant Infectious Diseases team is crucial to this end. Even if the candidate’s status as a carrier is associated with an increased risk of infection, this identification currently does not represent a contraindication to transplantation. However, it should warrant infection control measures, such as contact isolation precautions, and lead to a reasoned empirical antibiotic therapy in the case of infection [5,46].

### 2.4. Reasoned-Empirical Antibiotic Therapy in Early Intra-Abdominal Infections Post OLT and Source Control

Empiric antibiotic treatment post OLT needs to be tailored on microbiological isolates both from the donor (colonization and/or infection) and from the recipient (pre-OLT colonization). It also needs to consider the local epidemiology at the site where the transplant is performed [54].

#### 2.4.1. Donor Screening

Donor-derived infection may occur when an active infectious disease in the liver donor is not promptly recognized at the time of donation and/or correct pre-transplant bacterial screening is not performed. Adequate sampling of the donor (blood cultures and nasal and rectal swabs) is necessary to provide effective antibiotic prophylaxis in the recipient and decrease the risk of donor-derived bacterial infections [10].

In the case of an MDR bacterial colonization of the donor, the Transplant Infectious Diseases team should be immediately consulted to discuss the risks and benefits of transplantation and ensure appropriate antibiotic prophylaxis in the recipient [10].

#### 2.4.2. Recipient Screening

Recipient screening should be performed continuously, with nasal and rectal swabs, both at every pre-TOF outpatient assessment and at admission for transplantation. As discussed, colonization with MDROs is associated with increased mortality in liver transplant candidates, and there have been reports of differences in gut colonization MDROs during pre-OLT followup [55,56].

#### 2.4.3. Local Epidemiology

Active monitoring of local bacterial epidemiology, with up-to-date online data and feedback to the Transplant Infectious Diseases team, is critical to ensure adequate empiric antibiotic treatment, pending identification of the bacteria and antimicrobial susceptibility test (AST), in the case of suspected/proven post-OLT bacterial infection [57].

Therefore, a continuously updated knowledge of bacterial epidemiology in transplant wards and transplant ICUs is sorely needed. This can be challenging but is necessary especially in endemic areas for MDROs, such as ESBL Enterobacterales, KPC-producing *Klebsiella* spp., NDM-producing Enterobacterales, MDR *Pseudomonas aeruginosa*, MRSA, VRE, and carbapenem-resistant *Acinetobacter* spp. [57,58,59]. When a post-OLT abdominal infection is suspected, blood cultures and radiological imaging (e.g., ultrasound and/or CT scan) should be performed to exclude post-surgical abscess and the need for possible source control [17].

The choice of an empirical antibiotic regimen, if an infection is present, should always take into consideration not only the pathogens likely to be involved but other factors as well. In particular, the treating physician should always carefully evaluate the potential side-effect profile, drug–drug interaction, and the principle of antimicrobial stewardship [60,61]. Moreover, a rapid de-escalation should be performed as soon as microbiological data are available, to target the antibiotic therapy against the isolated organisms. As already discussed, the knowledge of colonization status may guide the choice of empiric therapy and reduce the time from sample collection to the start of appropriate in vitro active antibiotics.

The appropriate use of new antibiotics is important, especially in the very special population of transplant recipients. As a matter of fact, traditional antibiotics active against MDROs, such as colistin, may be counterindicated for these patients because of the high risk of side effects, usually nephrotoxicity. On the other hand, new antibiotics should be used appropriately in terms of indications, dosage, and duration.

A recent study conducted in solid organ transplant recipients with bloodstream infections caused by CRKP showed that patients treated with ceftazidime/avibactam (CAZ-AVI) had higher 14-day (80.7% vs. 60.6%, *p* = 0.011) and 30-day (83.1% vs. 60.6%, *p* = 0.004) clinical success and lower 30-day mortality (13.2% vs. 27.3%, *p* = 0.053) than those receiving the best available therapy (BAT) [62]. However, it should be considered that KPC-producing *Klebsiella pneumonia* with specific mechanisms conferring resistance to CAZ-AVI (such as D179Y variants) has been reported worldwide. In these cases, alternative regimens including meropenem/vaborbactam or imipenem/relebactam should be considered [63,64]. Again, the knowledge of local epidemiology is crucial, and an individualized approach is needed for the management of OLT recipients with MDRO infections after transplantation. Although the use of CAZ-AVI plus aztreonam has been associated with a reduced risk of death in patients with bloodstream infections by NDM-producing carbapenem-resistant Enterobacterales [65], no studies have been specifically conducted in OLT recipients.

Considering the increase in MDRO infections in this patient population, further studies are needed to evaluate the impact of MDRO infections on the outcome of OLT recipients and the best antibiotic treatment regimens in OLT recipients with MDRO infections. In a study of 331 adult post-OLT recipients at the University of Pittsburgh, deep surgical site infections (validated by culture) occurred in 15% of patients, at a median of 13 days post OLT [19]. These infections were characterized as abscesses/bilomas (58%), peritonitis (28%), deep incisional infections (8%), and cholangitis (6%). Enterobacterales (42%) and *Enterococcus* spp. (24%) were the predominant pathogens. Fifty-three percent of the bacteria were MDR, including 95% of *Enterococcus faecium* (VRE) and 55% of Enterobacterales (carbapenem-resistant and ESBL-producing organisms). Eighty-two percent of deep infections were caused by bacteria resistant to antimicrobials used for prophylaxis, and 58% of patients were treated with an inactive empiric regimen, resulting in increased 90-day mortality [19].

The following flowchart (Figure 1) shows the proposed empirical antibiotic treatment in the case of suspected post-OLT intra-abdominal infections by ESBL producing bacteria, and/or CRE and/or MRSA. Iif the presence of *Enterococcus* spp. is suspected or confirmed, an active antimicrobial agent against should be used.

Table 1 illustrates the most common antibiotics used and their usual dosage in patients with no renal dysfunction (eGFR ≥ 50 mL/min) or hepatic failure.

#### 2.4.4. Source Control

Appropriate source control is a major determinant of outcome in IAIs and can help shorten the course of antibiotic therapy. Source control is time dependent, as each hour of delay represents a negative factor in the outcome. Patients with IAIs should ideally undergo source control within 24 h [66]. Early source control in sepsis (<6 h) is associated with a reduced risk-adjusted odds of 90-day mortality [67].

The choice of the most appropriate procedure is determined by the operator experience, anatomical consideration, and clinical conditions (i.e., degree of critical illness and coagulopathy). For single small abscesses or small bilomas not associated with hepatic artery thrombosis, needle aspiration may be sufficient. On the other hand, for large collections, a percutaneous drainage catheter should be positioned and left in place until the drainage is minimal [68]. The optimal management of multiple collections or abscesses should be decided on an individual basis and requires the involvement of a multidisciplinary team (experienced transplant surgeons and physicians, interventional radiologists, and infectious diseases specialists). Multiple percutaneous interventions or laparotomy may be required [17,18]. Patients with diffuse peritonitis should undergo laparotomy as soon as possible. Direct aspiration of the peritoneal fluid or potentially infected collections should be sent for cultures to identify the responsible pathogens and to obtain the AST as soon as possible. Culturing the contents of indwelling catheters is not recommended as the microorganisms found in them usually represent drainage catheter colonization and may mislead the targeted antibiotic therapy [17].

## 3. Materials and Methods

A narrative review was carried out to retrieve the scientific evidence on bacterial infections occurring in adult liver transplant recipients within 1 month of transplantation.

A literature search using Pubmed was performed to select peer-reviewed articles published from 1 March 1967 (date of the first performed liver transplant) to 31 December 2022. The following search terms were used: “liver transplantation”, “hepatic transplantation”, “infection”, “bacterial infection”, “early”, and all the combinations of the abovementioned words.

In total, 471 records were found. After first screening by titles, 81 papers were retrieved. After abstract screening, five papers were excluded because they were in a language other than English. Of the remaining 76 papers, 68 (89.5%) were focused on early bacterial infection after liver transplantation in adults, while 8 (10.5%) were in pediatric patients. Thus, 68 papers were included in the study. The references of the selected studies were carefully assessed by two different authors (NR and RT) to identify articles not included in the primary search. The PRISMA 2020 flow diagram of the 471 identified papers is shown in Figure 2. The narrative review was organized in four sections: “Epidemiology and predisposing factors for IAI”; “Surgical-site infections and perioperative prophylaxis”; “MDRO colonization and infections”; “Reasoned-empirical antibiotic therapy in early intra-abdominal infections post OLT and source control”.

## 4. Conclusions

IAIs and surgical wound infections remain a common and significant problem after OLT, particularly among patients who experience surgical complications. These patients need to be monitored closely for post-transplant IAIs and for colonization by MDROs, which should warrant infection control measures and lead to a reasoned empirical antibiotic therapy in case of infection.

Multidisciplinary team communication between transplant surgeons, infectious disease specialists, microbiologists, and intensive care specialists is needed to individualize center-specific strategies for the prevention and treatment of IAIs in OLT, based on a patient’s risk factors, the operative factors, and the local microbiologic epidemiology.

The main limitation of our study is its intrinsic nature as a narrative review. However, our manuscript highlights in a practical way the main risk factors and subsequent appropriate therapeutic intervention in case of early intra-abdominal bacterial infections after OLT.

## Figures and Tables

**Figure 1 antibiotics-12-01316-f001:**
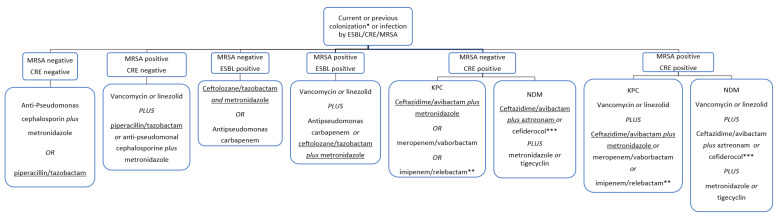
Reasoned-empirical antibiotic therapy flowchart in the case of post-OLT early abdominal infections. Suggested antibiotics can be considered in patients with no antibiotic allergies. If a Difficult-to-Treat (DTR) *P. aeruginosa* infection is suspected, avoid piperacillin/tazobactam and anti-pseudomonas cephalosporin, and prefer ceftolozane/tazobactam plus metronidazole or ceftazidime/avibactam plus metronidazole. * Colonization should be routinely tested with nasal swab for MRSA and rectal swab for ESBL and CRE. ** If the presence of *Enterococcus* spp. is suspected or confirmed, consider the use of imipenem for its anti-enterococcal activity. *** Considered only if the tested isolate shows a favorable MIC.

**Figure 2 antibiotics-12-01316-f002:**
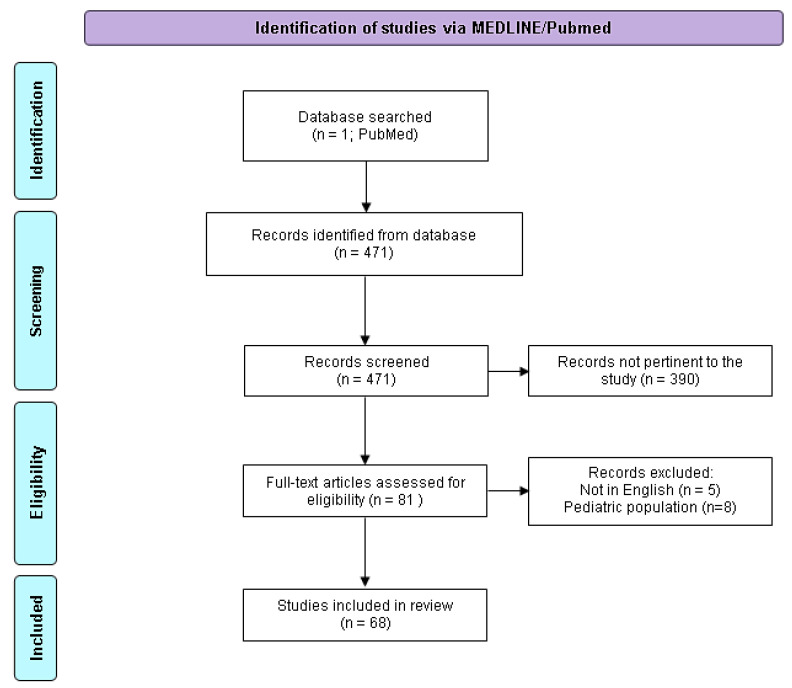
PRISMA 2020 flow diagram adapted for the present narrative review. Adapted from http://www.prisma-statement.org/ (accessed the 1 of May 2023).

**Table 1 antibiotics-12-01316-t001:** Antibiotics commonly used for reasoned-empirical antibiotic therapy in IAI post OLT.

Antibiotic	Dosage
Vancomycin	Loading dose 15–20 mg/kg intravenous (IV), then 30–40 mg/kg IV as continuous infusion
Linezolid	600 mg IV every 12 h or 1200 mg IV as continuous infusion
Tigecyclin	100 mg IV first dose followed by 50 mg IV every 12 h
Metronidazole	500 mg IV every 8 h
Meropenem	Loading dose 2 g IV and then 1 g every 8 h
Piperacillin/tazobactam	Loading dose 4.5 g IV and then 18 g as continuous infusion
Ceftolozane/tazobactam	1.5–3 g IV every 8 h
Ceftazidime/avibactam	2.5 g IV every 8 h *
Meropenem/vaborbactam	4 g IV every 8 h
Imipenem/relebactam	1.25 g IV every 6 h
Aztreonam	2 g IV every 8 h *

* Administer at the same time when NDM infection is suspected.

## Data Availability

Data are available after contacting the corresponding author.

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
