# Peer review of "Early Intra-Abdominal Bacterial Infections after Orthotopic Liver Transplantation: A Narrative Review for Clinicians"

_antibiotics, 2023, doi:10.3390/antibiotics12081316_

Round 1

Reviewer 1 Report

Manuscript ID: antibiotics-2539281

Title: Early intra-abdominal bacterial infections after orthotopic liver transplantation: a narrative review for clinicians.

Authors have written a review paper focused on bacterial intra-abdominal infections (IAIs) in orthotopic liver transplantation (OLT) of patients.

Authors need to revise whole manuscript thoroughly; in many places sentences are not clear and grammatically incorrect.  

Abstract must be improvised and written grammatically correct. Sentences should be written  short and self-sufficient to understand.

References must be rechecked in whole manuscript, is that appropriate for the mentioned text? Reference number 17 is missing in the manuscript.

Conclusion must be modified and rewritten. Conclusion started with limitation, it should be mentioned as a last sentence not first. Conclusion should be conclusive and in conclusion section there is no conclusive summary mentioned which can explain your study.    

 Others suggested comments mentioned below-

Line 3-Title should not have full stop sign at the end of text

Line 10-13- Abstract first sentence was not correctly written in the manuscript and was difficult to understand. Should rewrite the sentence and make it two sentences appropriately.

Line 16-18- Sentence written too long, it must be rewritten in two sentence.

Line 213-114- Sentence should rewrite in past tense.

This sentence is not clear “Multidrug resistance represents one of the most concerning issues worldwide and is 213 responsible for poor outcome in hospitalized patients.”

Line 360-61- In place of “antibiotic” must be mentioned “antibiotics”.

Figure 1 & 2 quality should be improvised and legend must be explained self-sufficient to understand.  

The authors need to revise the whole manuscript. English is good but it can be improvised. 

Author Response

Thank you for Your kind assessment of our manuscript .

We are here enclosing a point-by-point response to the comments, questions, and remarks that helped us improve the quality of the manuscript.

Comments:

Authors’ answers:

Authors have written a review paper focused on bacterial intra-abdominal infections (IAIs) in orthotopic liver transplantation (OLT) of patients.

Authors need to revise whole manuscript thoroughly; in many places sentences are not clear and grammatically incorrect. 

Abstract must be improvised and written grammatically correct. Sentences should be written  short and self-sufficient to understand.

We thank the Reviewer for the suggestions and we have grammar-checked the manuscript and updated the abstract.

References must be rechecked in whole manuscript, is that appropriate for the mentioned text? Reference number 17 is missing in the manuscript.

References have been double checked. Regarding reference 17, it is present in page 3, line 5 and it has been highlighted. Thank you for the concern.

Conclusion must be modified and rewritten. Conclusion started with limitation, it should be mentioned as a last sentence not first. Conclusion should be conclusive and in conclusion section there is no conclusive summary mentioned which can explain your study.   

We thank the Reviewer for the comment. We have updated the conclusion as suggested.

 Others suggested comments mentioned below-

Line 3-Title should not have full stop sign at the end of text

We have updated the title as suggested.

Line 10-13- Abstract first sentence was not correctly written in the manuscript and was difficult to understand. Should rewrite the sentence and make it two sentences appropriately.

We thank the Reviewer for the suggestion and we have rephrased the sentence as follow: “Despite recent advances in the transplant field, infectious complications after orthotopic liver transplantation (OLT) are major causes of morbidity and mortality”.

Line 16-18- Sentence written too long, it must be rewritten in two sentence.

We thank the Reviewer and we rephrased the sentence as follow: “We performed a narrative review of the literature in order to propose a “ready-to-use” flowchart for reasoned empirical antibiotic therapy in case of suspected post-OLT IAIs”.

Line 213-114- Sentence should rewrite in past tense.

We updated the sentence as suggested.

This sentence is not clear “Multidrug resistance represents one of the most concerning issues worldwide and is 213 responsible for poor outcome in hospitalized patients.”

We erased the sentence and rephrased the following one.

Line 360-61- In place of “antibiotic” must be mentioned “antibiotics”.

The sentence has been rephrased as follow: “antibiotic treatments”.

Figure 1 & 2 quality should be improvised and legend must be explained self-sufficient to understand. 

We thank the Reviewer for the suggestions and we have updated both figures .

Comments on the Quality of English Language

The authors need to revise the whole manuscript. English is good but it can be improvised.

The paper has been carefully revised by a native English speaker.

Reviewer 2 Report

Taddei et al. have submitted manuscript entitled "Early intra-abdominal bacterial infections after orthotopic liver 2 transplantation: a narrative review for clinicians." is written well. I have minor comments which may improve the manuscript quality

Author should work on overall structure of article. For instance, Lines 55-66 supposed to be one Paragraph. Is there any reason author divided them in multiple paragraph?

Moderate

Author Response

Thank you for Your kind assessment of our manuscript .

We are here enclosing a point-by-point response to the comments, questions, and remarks that helped us improve the quality of the manuscript.

Comments:

Authors’ answers:

Taddei et al. have submitted manuscript entitled "Early intra-abdominal bacterial infections after orthotopic liver 2 transplantation: a narrative review for clinicians." is written well. I have minor comments which may improve the manuscript quality

Author should work on overall structure of article. For instance, Lines 55-66 supposed to be one Paragraph. Is there any reason author divided them in multiple paragraph?

We thank the Reviewer for the kind assessment of our manuscript and we have revised the paragraph as suggested. Thank you once again.

Comments on the Quality of English Language

Moderate

The paper has been carefully revised by a native English speaker.

Reviewer 3 Report

“Early intra-abdominal bacterial infections after orthotopic liver transplantation: a narrative review for clinicians” was submitted to Antibiotics.

The manuscript deals with an interesting issue; however, there are several concerns related to the study.

In general terms, the manuscript is presented in a very disorderly manner that also requires an extensive revision of the grammar and language. After all the recommendations have been made, the possibility that the manuscript is relevant enough to recommend its publication will be evaluated.

Title: The title presents two relevant aspects that are not fully or adequately developed: “early intra-abdominal infections” and the "clinical" approach.

The authors must be coherent with the title, the objective, the methodology, the results, and the conclusions.

Abstract

The objective is totally different from the one described in the abstract.

A focus was placed on bacterial infections that occur within the first month from transplantation; an aspect that is not met.

In lines 16-18, other objectives are proposed that are not developed.

Lines 18-20. Some very general conclusions are described that do not account for the proposed objectives. It must respond to the objectives set.

Introduction

Line 16 of the abstract indicates the need to review recent literature; however, the introduction presents very outdated references.

The problem is described in a very weak and superficial form. The authors should present the importance of the review topic, the knowledge gap, the reasons for the clinical approach, and the importance of infection assessment during the first month.

The writing of the text must be fluid and orderly and not fragmented and incoherent with loose sentences without context (applies to the entire manuscript).

There are four subtitles (2 to 5) that do not understand what they correspond to. Are they the results? They should be introduced in the text to direct readers and not confuse them.

Point five and the figure seem to be the core of the results; however, they must be presented in a fluid and coherent manner. Moreover, it must be structured in such a way that a systematic order is generated that arouses interest in readers.

Results

As mentioned above this section is missing. Depending on the proposed objectives is recommended that the results be described in the same line.

Discussion

In the same way and in coherence, the results should be discussed. This segment is the most important part of the review; therefore, it is suggested to do it in a structured and coherent way. At the end of it, the limitations of the study should be presented.

M&M

This section also has major shortcomings. It is recommended that some questions be raised that allow the review to be developed in a coherent form. In view of the multiple objectives that the authors wish to develop, it is suggested that they propose some primary and secondary outcome variables. This will also allow you to generate an order in the manuscript.

Did you use Booleans in the search?

The first box in Figure 2 should be reviewed (n=1 ??).

Conclusions

The authors should focus their conclusions on the stated objectives.

Only 35% of the references are from the last 5 years. It is suggested that the review be as current as possible, especially in the introduction and discussion.

Extensive editing of English language required

Author Response

Thank you for Your kind assessment of our manuscript .

We are here enclosing a point-by-point response to the comments, questions, and remarks that helped us improve the quality of the manuscript.

Comments:

Authors’ answers:

The manuscript deals with an interesting issue; however, there are several concerns related to the study.

In general terms, the manuscript is presented in a very disorderly manner that also requires an extensive revision of the grammar and language. After all the recommendations have been made, the possibility that the manuscript is relevant enough to recommend its publication will be evaluated.

Title: The title presents two relevant aspects that are not fully or adequately developed: “early intra-abdominal infections” and the "clinical" approach.

The authors must be coherent with the title, the objective, the methodology, the results, and the conclusions.

 Abstract

The objective is totally different from the one described in the abstract.

We thank the Reviewer for the comment: we have completely rephrased the abstract.

A focus was placed on bacterial infections that occur within the first month from transplantation; an aspect that is not met.

Indeed, we have resumed the risk factors for early-abdominal bacterial infections after 1 month of OLT.

In lines 16-18, other objectives are proposed that are not developed.

Lines 18-20. Some very general conclusions are described that do not account for the proposed objectives. It must respond to the objectives set.

We thank the Reviewer for the comments: we have completely rephrased the abstract.

Introduction

Line 16 of the abstract indicates the need to review recent literature; however, the introduction presents very outdated references.

We thank the Reviewer for the comment, but the we meant to give to the Readers a comprehensive introduction that take in account also crucial references published 20 years ago. As a matter of fact, the oldest reference in the introduction has been published in 2006 (Transplantation – IF 6.2) .

The problem is described in a very weak and superficial form. The authors should present the importance of the review topic, the knowledge gap, the reasons for the clinical approach, and the importance of infection assessment during the first month.

The writing of the text must be fluid and orderly and not fragmented and incoherent with loose sentences without context (applies to the entire manuscript).

There are four subtitles (2 to 5) that do not understand what they correspond to. Are they the results? They should be introduced in the text to direct readers and not confuse them.

We changed the introduction according to the Reviewer suggestion and we specified in the Material and Methods section that the narrative review has been divided in 4 major paragraphs.

Point five and the figure seem to be the core of the results; however, they must be presented in a fluid and coherent manner. Moreover, it must be structured in such a way that a systematic order is generated that arouses interest in readers.

We thank the Reviewer for the comment and we have updated section 5 as suggested.

Results

As mentioned above this section is missing. Depending on the proposed objectives is recommended that the results be described in the same line.

Discussion

In the same way and in coherence, the results should be discussed. This segment is the most important part of the review; therefore, it is suggested to do it in a structured and coherent way. At the end of it, the limitations of the study should be presented.

We followed the indication of  “Instructions for Authors” for the Reviews (https://www.mdpi.com/journal/antibiotics/instructions).

 M&M

This section also has major shortcomings. It is recommended that some questions be raised that allow the review to be developed in a coherent form. In view of the multiple objectives that the authors wish to develop, it is suggested that they propose some primary and secondary outcome variables. This will also allow you to generate an order in the manuscript.

The Material and Methods section has been updated.

Did you use Booleans in the search?

No, Pubmed.

The first box in Figure 2 should be reviewed (n=1 ??).

It is correct. Pubmed is the only database used. So one (1).

 Conclusions

The authors should focus their conclusions on the stated objectives.

Only 35% of the references are from the last 5 years. It is suggested that the review be as current as possible, especially in the introduction and discussion.

The conclusion section has been updated.

Comments on the Quality of English Language

Extensive editing of English language required

The paper has been carefully revised by a native English speaker.

Reviewer 4 Report

This article investigated a critical issue: Early intra-abdominal bacterial infections after orthotopic liver transplantation. However, the manuscript has several issues that need to be addressed before considering it for publication:

Line 102: The abbreviations CT and MRI are mentioned here for the first time in this article. Therefore, they need to be spelled out and use only the abbreviations afterward.

Lines 129 – 132: It is mentioned here that “Liver transplant recipients have an increased 129 rate of SSIs compared with other solid organ transplants…” However, the reason behind this increased rate was not described. Did the appropriate administration of perioperative antibiotic prophylaxis affect the surgical site infection rate among the patient population in the studies that reported the rate difference?

Lines 220 – 222: CMV disease was described here as a risk factor for infections by resistant organisms. However, according to the timeline of infections after transplantation mentioned in lines 34 – 45, the CMV infection period (1-6 months from OLT) is after the bacterial infection period (First 30 days from OLT). Therefore, how can CMV disease be a risk factor for bacterial infections when it occurs after them?

Line 306: There is no need to spell the abbreviation “MDRO” again since it was already spelled out in line 123.

Figure 1: In the case of MRSA-negative/ESBL-positive infections, ceftolozane/tazobactam plus metronidazole was suggested as the first-line therapy. However, meropenem is an equally effective single-agent option that provides the same coverage and preserves the use of ceftolozane/tazobactam—perhaps making this option a second line needs to be explained.

Line 137: The article “the” is missing after the word “While” here.

Line 312: The word “adequate” was misspelled here.

Line 315: It is suggested to rewrite this sentence as follows for better understanding: “More challenging, but imperative, is the need for timely bacterial epidemiology…”

Lines 322 – 324: It is suggested to rewrite this sentence as follows for better understanding: “In case of infection, the choice of the empirical antibiotic regimen should also consider the potential side effect profile, the drug-drug interaction, and the principles of antimicrobial stewardship…”

Author Response

Thank you for Your kind assessment of our manuscript .

We are here enclosing a point-by-point response to the comments, questions, and remarks that helped us improve the quality of the manuscript.

Comments:

Authors’ answers:

This article investigated a critical issue: Early intra-abdominal bacterial infections after orthotopic liver transplantation. However, the manuscript has several issues that need to be addressed before considering it for publication:

Line 102: The abbreviations CT and MRI are mentioned here for the first time in this article. Therefore, they need to be spelled out and use only the abbreviations afterward.

We thank the Reviewer for the comment and we have added the complete spelling.

Lines 129 – 132: It is mentioned here that “Liver transplant recipients have an increased 129 rate of SSIs compared with other solid organ transplants…” However, the reason behind this increased rate was not described. Did the appropriate administration of perioperative antibiotic prophylaxis affect the surgical site infection rate among the patient population in the studies that reported the rate difference?

We thank the Reviewer for the comment, according to Anesi et al, when focusing on perioperative antibiotic prophylaxis no clear data are available for liver transplant, thus making the correct choice of prophylaxis difficult and less efficient when compared to pancreatic or renal transplant, for which stronger data on prophylaxis are available.

Lines 220 – 222: CMV disease was described here as a risk factor for infections by resistant organisms. However, according to the timeline of infections after transplantation mentioned in lines 34 – 45, the CMV infection period (1-6 months from OLT) is after the bacterial infection period (First 30 days from OLT). Therefore, how can CMV disease be a risk factor for bacterial infections when it occurs after them?

We thank the Reviewer for the comment and we removed the sentence.

Line 306: There is no need to spell the abbreviation “MDRO” again since it was already spelled out in line 123.

We thank the Reviewer for the comment and we removed the repetition

Figure 1: In the case of MRSA-negative/ESBL-positive infections, ceftolozane/tazobactam plus metronidazole was suggested as the first-line therapy. However, meropenem is an equally effective single-agent option that provides the same coverage and preserves the use of ceftolozane/tazobactam—perhaps making this option a second line needs to be explained.

We thank the Reviewer for the comment. In highly endemic countries for carbapenem resistant MDRO, carbapenem sparing strategies should be preferred over the systematic use of antipseudomonas carbapenem to decrease the incidence of resistance. Thus, we suggest, when feasible both in terms of resistance and costs, ceftolozane/tazobactam plus metronidazole over carbapenems.

Comments on the Quality of English Language

Line 137: The article “the” is missing after the word “While” here.

Line 312: The word “adequate” was misspelled here.

Line 315: It is suggested to rewrite this sentence as follows for better understanding: “More challenging, but imperative, is the need for timely bacterial epidemiology…”

Lines 322 – 324: It is suggested to rewrite this sentence as follows for better understanding: “In case of infection, the choice of the empirical antibiotic regimen should also consider the potential side effect profile, the drug-drug interaction, and the principles of antimicrobial stewardship…”

We thank the Reviewer for the comments and we revised all indicated sentences.

Round 2

Reviewer 3 Report

Although the authors improved some aspects of the manuscript, other recommendations were not considered. Therefore, the authors must adjust the following aspects:

Abstract

-The objective is totally different from the one described in the abstract.

-A focus was placed on bacterial infections that occur within the first month from transplantation; an aspect that is not met.

-Some very general conclusions are described that do not account for the proposed objectives. It must respond to the objectives set.

Subtitles 2 to 5 should be developed under a title that indicates that they are the results. This will guide the readers properly.

Discussion

The instructions to the authors indicate that this section is mandatory; however, this section was not carried out in the manuscript. This segment is the most important part of the review; therefore, it is suggested to do it in a structured and coherent way. At the end of it, the limitations of the study should be presented. This revision is full of limitations that must be recognized. Moreover, it is not usual to place limitations at the end of the conclusions.

M&M

This section also has major shortcomings. It is recommended that some questions be raised that allow the review to be developed in a coherent form. In view of the multiple objectives that the authors wish to develop, it is suggested that they propose some primary and secondary outcome variables. This will also allow you to generate an order in the manuscript.

THE AUTHORS DID NOT RESPOND TO THIS REQUEST. THIS IS A VERY IMPORTANT ASPECT THAT MUST BE DEVELOPED.

The first box in Figure 2 should be reviewed. Database searched= PubMed.

Did you use Booleans in the search?

The use of Boolean connectors between keywords is a very important strategy in the search for scientific literature. Therefore, this is another of the shortcomings of this review.

It would be appropriate to include the type of articles included in the review and their level of evidence according to the design used.

minor editing

Author Response

Thank you once again for your kind assessment of our manuscript.

We are here enclosing a point-by-point response to the Reviewer’s comments, questions, and remarks.

Comments:

Authors’ answers:

Abstract

-The objective is totally different from the one described in the abstract.

-A focus was placed on bacterial infections that occur within the first month from transplantation; an aspect that is not met.

-Some very general conclusions are described that do not account for the proposed objectives. It must respond to the objectives set.

We thank the Reviewer for the comment. The abstract has been updated as follow: “Despite recent advances in the transplant field, infectious complications after orthotopic liver transplantation (OLT) are major causes of morbidity and mortality. Bacterial intra-abdominal infections (IAIs) are predominant during the first month post-transplantation and affect patient and graft survival. Recently, the emergence of multi-drug resistant bacteria is generating great concern in OLT patients. We performed this narrative review of the literature in order to propose a “ready-to-use” flowchart for reasoned empirical antibiotic therapy in case of suspected post-OLT IAIs. The review was ultimately organized in four major chapters: “Epidemiology and predisposing factors for IAI”; “ Surgical site infections and perioperative prophylaxis”;  “MDRO colonization and infections”; “Reasoned-empirical antibiotic therapy in early intra-abdominal infections post OLT and source control”. A multidisciplinary team work is warranted to individualize strategies for prevention and treatment of IAIs in OLT recipients, taking in account patient’s risk factors, surgical characteristics and local bacterial epidemiology”

Subtitles 2 to 5 should be developed under a title that indicates that they are the results. This will guide the readers properly.

We thank the Reviewer for the suggestion and we have enclosed in the paper a Results section.

Discussion

The instructions to the authors indicate that this section is mandatory; however, this section was not carried out in the manuscript. This segment is the most important part of the review; therefore, it is suggested to do it in a structured and coherent way. At the end of it, the limitations of the study should be presented. This revision is full of limitations that must be recognized. Moreover, it is not usual to place limitations at the end of the conclusions.

As indicated in the guidelines of the Journal for the Review article: “The structure can include an Abstract, Keywords, Introduction, Relevant Sections, Discussion, Conclusions, and Future Directions”. We preferred to keep the structure coherent with the nature of the Review that aims to give Clinicians worldwide an easy to use tool for empiric antibiotic treatment in post-TOF early-abdominal  infections.

Moreover, Reviewer 1 esplicitly asked us to put limitations at the end of the conclusion section.

M&M

This section also has major shortcomings. It is recommended that some questions be raised that allow the review to be developed in a coherent form. In view of the multiple objectives that the authors wish to develop, it is suggested that they propose some primary and secondary outcome variables. This will also allow you to generate an order in the manuscript.

As a narrative review, outcomes variables are not indicated.

THE AUTHORS DID NOT RESPOND TO THIS REQUEST. THIS IS A VERY IMPORTANT ASPECT THAT MUST BE DEVELOPED.

The first box in Figure 2 should be reviewed. Database searched= PubMed.

 We thank the Reviewer for the comment. The text in the figure has been modified according to the suggestion.

Did you use Booleans in the search?

The use of Boolean connectors between keywords is a very important strategy in the search for scientific literature. Therefore, this is another of the shortcomings of this review.

We underlined in the M&M section that:” A literature search using Pubmed was performed to select peer-reviewed articles published from 1/Mar/1967 (date of the first performed liver transplant) to 31/Dec/2022. The following search terms were used: “liver transplantation”, “haepatic transplantation”, “hepatic transplantation”, “infection”, “bacterial infection”, “early”, and all the combinations of the above-mentioned words”.

It would be appropriate to include the type of articles included in the review and their level of evidence according to the design used.

The full excel list of the articles is at disposal of the Reviewer.
